# Protocol Article: A Cross-Sectional Evaluation of Children’s Feet and Lower Extremities

**DOI:** 10.3390/mps6060115

**Published:** 2023-12-01

**Authors:** Christian Wong, Christina Ystrøm Bjerge, Ales Jurca, Michael Mørk Petersen, Soren Boedtker, Andreas Balslev-Clausen, Steen Harsted

**Affiliations:** 1Department of Orthopedic Surgery, Copenhagen University Hospital, 2650 Hvidovre, Denmarkandreas.peter.balslev-clausen.02@regionh.dk (A.B.-C.); 2Department of Orthopedic Surgery, Copenhagen University Hospital, Rigshospitalet, 2100 Copenhagen, Denmark; michael.moerk.petersen@regionh.dk; 3The Association of Danish Podiatrists, 2300 Copenhagen, Denmark; cb@fodterapeut.dk; 4Volumental AB, 118 72 Stockholm, Sweden; ales.jurca@volumental.com; 5Jozef Stefan International Postgraduate School, 1000 Ljubljana, Slovenia; 6Center for Muscle and Joint Health, Department of Sports Science and Clinical Biomechanics, University of Southern Denmark, 5230 Odense, Denmark; sharsted@health.sdu.dk; 7Medical Research Unit, Spine Center of Southern Denmark, University Hospital of Southern Denmark, 5500 Middelfart, Denmark

**Keywords:** anthropometrical measurements, lower extremity, foot, foot pathologies, socioeconomic factors, sports activities, pain status, children, protocol article

## Abstract

Background: The health of children’s lower extremities and feet is a focus area for caregivers and healthcare professionals such as doctors, school nurses, and podiatrists. Our study aims to investigate the general health status of Danish children’s lower extremities and feet to identify anthropometric parameters that might be preconditions for pain and evaluate for foot diseases and whether they are associated with pain intensity and location, three-dimensional foot dimensions and foot pressure mapping, shoe dimensions, types and intensity of sports activity, quality of life, and foot health. The aim is that we will be able to identify parameters pre-dispositioning for pain, thus providing recommendations for sports activities in relation to the anthropometric conditions of a child as a potential preventive measure for pain. This analysis will be stratified by socioeconomic status on a group level, and this perspective will be able to provide preventative recommendations to prevent pain. Methods: This study is a cross-sectional examination of a thousand children in the first, fifth, and ninth grades in randomized selected Danish primary schools. We will perform a clinical examination of the lower extremities and feet for misalignments, deformities, and diseases as well as rotational status and range of motion. Moreover, we will evaluate their pain levels, sports activities, three-dimensional foot dimensions, plantar pressure, footwear, and patient-related outcome measures (PROMs) for foot health and quality of life. Results: We aim to provide an anthropometrical overview of the lower extremities and feet in children. The obtained basic understanding of healthy normal material in children will be analyzed for its relationships with pain level, sports activities, and socioeconomic status on a group level. This could potentially provide us with an understanding of the factors that impact lower extremity and foot diseases in children. In conclusion, examining children’s lower extremities and feet in Danish primary schools is a step toward identifying areas of improvement in self-care and shoe fitting, mapping podiatry-related needs of care in children’s feet, and providing parental recommendations for preventive actions on shoe fitting and the choice and intensity of sports activity concerning pain. Conclusions: The tenet of this study is a long-term follow-up to evaluate the long-term socioeconomic course on a group level, foot status, and sports activity, using patient-related outcome measures evaluating quality of life and other lifestyle factors such as emotional functioning, social functioning and interaction, and school functioning. Potentially, this will improve children’s quality of life and prevent future diseases.

## 1. Introduction

The well-being of children, and the health of children’s feet, is a priority for a range of professionals, such as school nurses, pediatricians, general practitioners, pediatric orthopedics, and podiatrists [1,2]. Social inequality impacts general health in an environment of growing social differences in an increasingly fragmented healthcare system and might raise barriers to accessing qualified health care, thus indicating a need for either rehabilitation and preventive actions or information on healthcare issues concerning children’s diseases of the lower extremities and feet, and a study evaluating the need of healthcare services indicates that this might not reach medical consultation [3,4,5,6,7]. Healthy feet are a prerequisite for many physical activities, and painful feet may inhibit children in physical exercise and give a decreased quality of life [8,9,10]. Children can have painful feet for a plethora of reasons, and structural differences may have an impact on pain and level of activity. Parents and grandparents are concerned that children with in-toeing, flat feet, and curly toes may develop pain or disability, and healthcare workers must be able to differentiate between what are merely physiological variations and what may lead to detrimental conditions as a child grows older. Previous studies indicate that children are born with “flat feet’’ [11,12,13,14] and that the longitudinal medial arch is not fully developed until a child is 6–7 years old [15,16]. Therefore, several studies have failed to show that insoles and special shoes in preschool children change the natural history of flat feet [17]. However, studies examining anthropometric parameters of the lower extremities and feet have not been performed in older children, and little is known about how this affects an older child in relation to pain and how this may affect their participation in sports and other physical activities. Children with flat feet may have a reduced quality of life, thus motivating interventions. A recent study indicates that insoles in older children may change the osseous construction of the foot and thus, may prevent painful flat feet in adults [18]. Therefore, insoles may have the potential to prevent painful flat feet in adults and potentially prevent surgery for this condition in adults [19]. For these motivating reasons, the pediatric orthopedics section of the Rigshospitalet and Hvidovre Hospital, the University of Southern Denmark, and the Association of Danish Podiatrists have started a project for the evaluation and screening of children’s feet and lower extremities in Danish primary schools. Wearing shoes can be underestimated concerning the health of children’s feet, but a too-tight-fitted shoe is known to predispose to hallux valgus and lead to foot pain in children [20,21,22]. Studies in the European setting indicate that children in these countries often wear shoes that are too small [23,24]. The effect of shoe fit also impacts the development of pathological diseases such as apophysitis of the calcanei, and simple ingrown toenails and may give rise to deformities, ingrown toenails, or non-specific foot pain [20,22,25]. In this study, we plan to evaluate this using the commercial technology of 3D surface scanning and evaluate the appropriateness of shoe fitting via the measurement of shoe size. The perspective of this study is that we, in the future, will be able to provide recommendations for preventive actions for children for improvement in the care of the health of children’s lower extremities and feet, thus demonstrating the potential for long-term health and societal financial benefits, such as referral to other health services, guidance on sensible footwear, foot care, and hygiene. The purpose of this project is to identify anthropometric parameters that might be preconditions for pain, evaluate for foot diseases and whether they are associated with pain intensity and location, three-dimensional foot dimensions and foot pressure mapping, shoe dimensions, intensity and type of sports activity, and status of quality of life and foot health according to patient-related outcome measures (PROMs). The aim is that we will be able to identify parameters pre-dispositioning for pain, thus providing recommendations for sports activities in relation to the anthropometric conditions of a child as a potential preventive measure for pain. Our purpose with this study is threefold:To make reference material of the demography of the foot, the rotational status, and the range of motion in joints in the lower extremities of children in Denmark to evaluate if these anthropometric parameters in children influence pain status.Examining the prevalence of pain in children as well as exploring the prevalence of pathologies in children’s feet to evaluate if these conditions in children influence pain status.Evaluating the appropriateness of shoe fitting using commercial 3D technology and foot pressure mapping to evaluate if shoe fitting in children influences foot pain status.

This will be examined from the perspective of detecting and clarifying the role of sports activities, pain status, and socio-economic inequalities on a group level in the health of Danish children’s feet.

## 2. Method and Materials

### 2.1. Study Design

The design of this study is a cross-sectional examination or a clinical and observational study of children in the first, fifth, and ninth grades in Danish primary schools. The schools will be selected randomly by their socioeconomic profiles. Our focus will be on analyzing their clinical status, pain levels, foot dimensions and pressure, and footwear. Specifically, we plan to evaluate misalignments, malpositions, deformities, skin and foot hygiene, and the rotational status of the lower leg, as well as the range of motion in the lower extremities of the children. These data will serve as a reference for the foot demographics and provide insights into the health of children’s feet. We will enquire about foot health and quality of life using PROMs, and we will measure foot dimensions using 3D scanning techniques on the children’s feet and assess the quality of their footwear by evaluating shoe sizes with scientific measurements. This will help us determine if there is a mismatch between footwear and foot size, which could potentially lead to deteriorating foot health in children. Additionally, we plan to collect data on foot pressure mapping to determine the pressure points on the children’s feet. We will evaluate their pain status based on the anatomical pain location, type, and intensity using numerical pain rating scales. 

#### 2.1.1. Study Plan

This study was initiated on 1 April 2022 and ends on 1 May 2025 (end of the project). We planned for the last visit and last subject to be on 1 January 2023. However, due to COVID, the project has been delayed. 

#### 2.1.2. Study Organization

This study is organized under the leadership of a senior medical consultant and one senior researcher who oversees the employment and supervision of a team of researchers. The team consists of three medical students (TAP), four doctors, six podiatrists, one project coordinating nurse, and one senior researcher as a collaborator. Regularly scheduled meetings are held throughout the school year, with ad hoc meetings as necessary to ensure effective coordination and collaboration among team members. This study will be conducted in collaboration with 20 schools at different locations in Region Hovedstaden and Region Sjaelland. 

## 3. Participants

A thousand children from the first, fifth, and ninth grades in Danish primary schools will be asked to participate in this study.

### 3.1. Inclusion Process

When a child enters this study, we will not intervene in any prescribed therapy and medication. We plan to include a thousand children (with a possible dropout rate of 5%), which will be possible due to a collaboration with a selected number of schools in Region Hovedstaden and Sjaelland. All participants will be recruited through the director of the participating school. One local co-investigator (not associated with this study) will be appointed from each school. The inclusion process will continue until a total of a thousand subjects are included in this study. The project coordinator and the co-investigator at each school will send out a specific link to relevant information for the possible candidates and obtain a signed informed consent. This link will be mediated via the Danish parental-school communication channel, the Aula portal. Access to Aula will be through parents or guardians who must log in with the Danish national identification service NemID. The data, which are mediated, are stored in Europe. Aula is hosted by Amazon Web Services (AWS). AWS is an American-owned company that provides cloud solutions and has data centers in Ireland, Germany, and France, among others. However, part of the Aula agreement is that data must be located in an EU/EEA country. Data in the Aula system are encrypted. Data are encrypted throughout the process. The keys to the encryption are securely stored in Denmark, and Aula complies with the GDPR rules and rules regarding IT auditing. This is mediated through a previously approved procedure in Region H for the Redcap database and a similar procedure has been approved in a project in ‘De Videnskabsetiske Komitéer for Region Midtjylland’. As mentioned, all participants will then be recruited using a method that is approved by the local information security service CIMT, region H, Copenhagen Denmark*, where we will obtain consent for a research project using the REDCap system by asking the subjects and caregivers for consent to participate in a project using the following procedure: The subject and caregiver log on to Aula with NemID.A link to RedCap is sent via Aula to the subject and caregiver (who has now been verified via the NemID login).The subject and caregiver enter RedCap to be able to assess written patient information and preliminary oral information via video, with the possibility to orally inquire about additional information on the project (see below).After this, they will be able to confirm his/her consent by signing into the digital system.

This method is applied since the Section for Information Security* assesses that the verification of the citizen that happens when the subject and caregiver must log into the e-box with their NemID to access RedCap and give his/her consent is in itself sufficient. The Section for Information Security, therefore, considers the subsequent signature with a digital device as an “extra” measure that is not decisive for the actual verification. The Section for Information Security can thus confirm that this method of obtaining consent via RedCap is legally compliant in terms of data protection law and that the method in the Capital Region is “approved”. Information will be sent to Redcap as described. Potential candidates for this study will be identified and screened for inclusion. Screening and assessments will be performed by the project coordinator and project leader to determine eligibility. As the participants in this study are minors, informed consent from both parents will be obtained after receiving written information about the study from the project coordinator as described. Written and oral information via video and the project coordinator, which will initially be provided via Redcap, will be sent/administered. The child and caregiver will be given the possibility to contact the investigators via email and phone for further information about the project and relevant information to possible candidates before giving their consent. The project coordinator will make sure that participation in this study also follows the interests of the child. Should the parents have any questions regarding this study, they can contact the project coordinator via mail or phone. If parents and/or their child(ren) do not want to participate, they can withdraw at any time during this study, thus following the local guidelines of informed consent. We will include the subjects according to the following criteria:

### 3.2. Inclusion Criteria

Children aged 6–18, and the children must attend 1st, 5th, or 9th grade at a Danish primary school.Children from selected schools in Region Hovedstaden and Region Sjælland.

### 3.3. Exclusion Criteria

Not accepting to participate in this study.Malignancies or infections discovered throughout this study that warrant acute medical treatment.

### 3.4. Treatment during and after This Study Has Ended/Exclusion from This Study and Discontinuation of This Study

If a child is discovered to have severe pathologies such as malignancies of infection that need acute intervention during this study, then the child has to be excluded and treated. After exclusion, the child and their parents have the possibility of us sharing the discovered information (from this study) about their child’s health status as well as if we will recommend seeking medical care. If a child is being treated, we will allow the child to continue being treated. If a child participating in this study has an unexpected negative effect due to the evaluation, either according to the parents or based on the measurements performed, an assessment by the project coordinator (in collaboration with the parents) will be carried out to decide if the child should be excluded. The reason for exclusion will be reported, and it will be decided if it is still safe to continue the study. The child will continue to receive the best possible care when entering this study. 

### 3.5. Insurance

Upon participating in this study, the participants are covered by the Danish patient insurance “Patientforsikringen” and can complain to “Patientklagenævnet” if unsatisfied with this study.

## 4. Ethical Consideration

### Ethics Approval and Consent to Participate

Before the commencement of the project, we applied to the local ethics committee for ethical approval (H-22002997). The study design was evaluated as a cross-sectional non-intervention study, thus not appropriate for evaluation for approval in accordance with local legal law § 2 or registration as a clinical trial. However, we can confirm that all methods and procedures of this study will adhere to relevant national guidelines and regulations. This includes obtaining informed signed consent from all subjects’ legal guardian(s). If subjects and/or the legal guardians refuse to participate, this will be accepted. Regional registration was obtained in accordance with the Danish Data Protection Agency, as stipulated by Danish law J.nr. 2008-41-2240. 5.3 Ethical Considerations.

The clinical evaluation of children will be a standard orthopedic examination. In our experience, this examination does not cause pain or have side effects/complications, and we expect that the children will not experience pain during the examination, except for minor discomfort. This study entails the evaluation and surface scanning of the children’s legs and feet. All procedures are painless and well-known without known potential side effects or inducing radiation via X-rays. Participants in this study will not experience an increase in the risk of complications or side effects. This study will provide normal materials for orthopedic lower extremity evaluation in children. Previous normal materials for such evaluations are based on a smaller international dataset from decades ago, as described earlier. There is a need to explore this further since the demography has changed during the last decades, and the inclusion of children in this study is justified. We believe that children participating in this study, in addition, will gain information about their anthropologic data as well as their foot and pain statuses. Thus, we believe that the benefits outweigh the risks.

## 5. Procedure and Method

### 5.1. Assessments

In this study, we will utilize clinical evaluations, computer-based technology, and PROMs to gain a more holistic understanding of children’s foot problems and their impact on quality of life. 

### 5.2. Clinical Examinations

The clinical examination in this study is based on the prevalence of pathologies in children to obtain data on the normal material of these children. This entails the examination and registration of the following: -Mapping the foot anatomy and pathologies of warts, foot deformities, ingrown toenails, callosities, hallux valgus, metatarsus varus, and the ROM of the subtalar joint and midfoot, hindfoot valgus, navicular height, static foot pressure mapping, and foot type (square, Roman, or Greek) [26].-Mapping the anthropometric data on children’s height and weight, hip range of motion (ROM) (flexion, internal and external ROM [27], the ROM of knee flexion, and the extension ROM of the ankle joint on a flexed and extended knee [27], as well as the femoral anteversion and torsion of the tibia (foot–thigh ankle)). This is evaluated when the child is either standing, prone, or supine in a standardized manner and using a goniometer (see Appendix A).

### 5.3. Patient-Related Outcome Measures (PROMs) and Pain Status

Our endpoints for this study are foot health, pain status, sports activity level, and quality of life. We will evaluate pain intensity and location using the ‘Chili’ score, Wong–Baker and VAS numeric rating scale, and a systematic pain drawing, respectively [28]. We will measure foot health using the validated Danish version of the Oxford Foot Score for Children (OxAFQ) and compare scores with the prevalence of foot pathologies [29]. The OxAFQ is a discriminative and evaluative tool that can differentiate between severe chronic disease and benign fluctuating disease. We will assess activity levels and the Pediatric Quality of Life (PedsQL) questionnaire. The 23-item PedsQL Generic Core Scales are designed to measure the core dimensions of health as delineated by the World Health Organization. These data will be acquired either via REDCAP via a subject’s social security number for the questionnaires, and pain status will be evaluated when a child is examined. The caregiver and child will also be asked if the child has orthopedic and other diseases.

### 5.4. Equipment

We will evaluate the dimensions of a child’s feet [30,31,32] with a 3D scan using the Volumental scanner (Volumental, Stockholm, Sweden) in a neutral stance and when standing on their toes and with foot pressure mapping [33]. The mapping of the quality of footwear and the size of shoes will be evaluated with measurements of shoe width and length using the MFI shoe evaluation tools [30,32,34]. Figure 1 illustrates the 3D scans, foot pressure mapping, and shoe evaluations.

#### 5.4.1. Measurement Procedure for the Foot

The Volumental 3D foot scanner will be used to scan both feet of a child. Eleven foot-anthropometry parameters will be measured: (1) foot length, (2) forefoot width, (3) arch height, (4) instep height, (5) heel width, (6) ball girth, (7) instep girth, (8) ankle wrap, (9) short heel girth, (10) forefoot height, and (11) hallux valgus angle. The Volumental 3D foot scanner and the anchor point of measurement for each dimension can be seen in Figure 1a,b, respectively. The procedure of the 3D foot scan requires that the participants are barefooted during measurement with a time measurement for each subject of approximately three minutes. The examinators use a tablet to interact with the scanner during the foot-scanning process and to show the scan results to the child. The child needs to take off their shoes and socks before scanning and be wearing shorts that are purchased for this study. The child stands on the base platform of the scanner with both feet positioned slightly apart from one another. Bodyweight should be equally distributed between both feet. We will also acquire scans when the child is standing on their toes. The researcher selects the sex and starts the scan. To capture both feet, the scanning process takes 5 s. After the scan is successfully processed, 3D meshes and foot measurements of both feet are displayed to the child. 

#### 5.4.2. Data Collection Footwear

Data are collected from the outdoor shoes that the children wore to preschool on the day of the examination. Parents are not informed ahead of time on which days data will be collected. 

##### Shoe Length

The inside length of the outdoor shoes is measured using an adapted sliding device designed for this purpose Figure 2a. To determine the inside length with a maximum level of precision, we utilize a commercially available In-Shoe Length Measuring Device (LM, 3079/1/2/3/4, reference N° LM 3074/1/2/3). This hand-held device determines the in-shoe length and, hence, the insole length of shoes by measuring the different heel and toe spring values. The device is seen in Figure 2a.

Measurements are conducted on the insole at one-millimeter intervals. The expansion of the measuring device lengthwise is determined by the resistance generated by the material of the heel counter and the reinforcement of the toe tip. We plan to calculate the lengthwise fit of the outdoor shoes by the difference between the actual inside length of the shoe and the length of the foot. The shoe should be at least 10 mm longer than the foot when classifying the shoe as a properly fitting shoe [35].

##### Shoe Width

The width of the footwear will be measured from the widest point on the inside of the shoe by measuring the sole. A measuring device is designed to determine the maximum shoe width and height at the level of the first metatarsal head (FH). To determine the inside length with a maximum level of precision, we utilize a commercially available In-Shoe Width Measuring Device (SWM 3079/1/2, reference N° SWM 3079/1 = adults SWM 3079/2 = children). This hand-held device determines the shoe (joint) width and girth of the completed footwear. Using a loop-type elastic band, the gauge is inserted and fitted to the shoe. The gauge has a brake that can be released once inside the footwear, allowing the instrument to expand transversally and upward until it meets the inner edges of the shoe. This will activate a break and when extracting the gauge, we will determine the width and height of the footwear. The device is seen in Figure 2a.

#### 5.4.3. Foot Pressure Mapping

We will measure one postural static foot pressure mapping using the Sidas pressure posturology platform presscam V5 (SIDAS SAS, Voiron, France). This scanner is a hybrid pressure posturology platform and is powered via a USB cable connected to a computer. This computerized system consists of a floor mat on which an ultra-thin Tekscan sensor of 960 sensor cells (5 mm2 each) is placed. The mat will be calibrated for each child using the child’s weight before each testing session. The child stands barefoot on the mat, and the maximum plantar pressure for the entire foot is obtained in 300 images. Measurement of the total surface area of both feet, the maximal pressure on both feet (g/cm^2^), and the average pressure on both feet will be required. This will allow us to acquire plantar pressure data in the static mode and center of the pressure center. The device is seen in Figure 2b.

## 6. Analyses and Statistics

The design of our study is a cross-sectional examination of a thousand children from the first, fifth, and ninth grades in Danish primary schools. These will be randomly selected by their socio-economic profiles. Our focus will be on analyzing their clinical status, pain levels, foot dimensions, pressure, and footwear. Specifically, we plan to evaluate misalignments, malpositions, deformities, skin and foot hygiene, and the rotational status of the lower leg, as well as the range of motion in the lower extremities of the children. These data will serve as a reference for the demographic of the foot and provide insights into the health of children’s feet. We will measure foot dimensions using 3D scanning techniques on the children’s feet and assess the quality of their footwear by evaluating shoe sizes with scientific measurements. This will help us determine if there is a mismatch between footwear and foot size, which could potentially lead to deteriorating foot health in children. Additionally, we plan to collect data on foot pressure mapping to determine the pressure points on the children’s feet. We will evaluate their pain status based on the anatomical pain location and type using numerical pain rating scales. 

### 6.1. Availability of Data and Materials

The data in this study will be registered and reported in a REDCap database following the GDPR rules of the Danish data protection agency and will be stored for 5 years after this study has ended. The data of each subject will be filled out by the examining podiatrists assisted by the TAP employee directly on paper, which afterward, will be transferred to the REDCAP database by TAP employees with secure internet access to register/record the data of this study directly in the REDCAP database as well as for any of the apparatus used in this study. General information about the children will be collected from the primary caregiver of each patient using the methodology via Aula and REDCap as described in Section 2.1.

All relevant information collected during this study will be passed on to the participating doctors for evaluation, if necessary. Information regarding the patients will be passed on directly to the participating doctors, if necessary, by using “…@regionh.dk” emails or directly, using the REDCAP database to ensure the protection of the data. All transfers of data will comply with and have been approved according to the current GPDR rules. 

All data about the patients will appear pseudo-anonymized using the patients’ ID numbers. Information will be kept on a logged and secure drive in the RegionH intranet system. Information from medical records will not be retrieved, but data to provide information about this study to the parents and inquire about informed consent will be required to contact them by letter, as described in Section 2.1. This information includes name, age, CPR number, gender, address, and email. All information regarding the patients will be protected according to “Lov om behandling af personoplysninger og Sundhedsloven, afsnit 3 ver. patienters retsstilling”. Information regarding this study, and, thus, the patients, will also be passed on to governmental agencies (health authorities and the Danish Data Protection Agency if an inspection of this study is requested). Data will not be transferred to other countries. The dataset for this study has been generated and is under preliminary analyses and is thus not currently publicly available. For further interest, inquiries about the data can be sent to the corresponding author.

### 6.2. Safety Endpoints and Evaluations

This study is initiated by the Orthopediatric Department at Hvidovre Hospital and will be conducted in collaboration with 20 schools at different locations in Region Hovedstaden and Region Sjælland. This study will be evaluated by the Danish Health Authorities, The National Committee on Health Research Ethics, and the Danish Data Protection Agency (Privacy in RegionH). This study will be conducted according to the guidelines described in this protocol. This study (as in any study) will be monitored for adverse reactions, adverse events, serious adverse events, serious adverse drug reactions, and suspected unexpected, serious adverse reactions following the ICH GCP harmonized guidelines, and it will be estimated by the project coordinator if any event that has occurred could be attributable to the test measurements. This will be assessed continuously. The project coordinator will ask the child and parents in case an event has occurred. The parents will be asked to call the hospital if any serious event has occurred. The treating physician will register and categorize the event in the REDCAP database and report the event to the relevant authorities. The categorization of the event will be performed using the investigator’s brochure as a reference. In this study, adverse reactions and adverse events are defined as follows according to the ICH GCP guidelines. 

### 6.3. Statistical Analysis and Randomization 

Three clusters of equally divided children from the 1st grade, 5th grade, and 9th grade will be evaluated. This was chosen to embrace the spectrum of chronological (with different developmental stages) ages in Danish primary schools. The total planned number of included children is a thousand children. The sample size was chosen based on a balance between the desired statistical robustness and the practical constraints of this study. These constraints include available resources, such as funding, staffing, and logistics. This sample size is in line with or larger than similar large-scale, observational studies in pediatric populations [36,37,38], where specific power calculations can be more complex due to the multitude of variables and outcomes being explored. We acknowledge this as a limitation but also believe that the chosen sample size is sufficient to provide meaningful and robust data for the objectives of our study. The schools will be chosen randomly with the measure of crow fly distance and the number of children in the schools (>1000). A school’s socioeconomic status will be registered based on the ‘Socioøkonomisk reference for grundskolekarakterer 2018/2019′ [36]. The normality of the data will be assessed graphically using quantile–quantile and density plots. 

Based on their distribution, variables will be described using either means and standard deviations (for normally distributed data) or medians and interquartile ranges (for non-normally distributed data). Appropriate hypothesis tests will be conducted based on the distribution of the data. These include Student’s t-test and Wilcoxon’s rank-sum test for continuous data. For categorical data, chi-square tests and Fisher’s exact test will be used as appropriate. *p*-values of ≤ 0.05 will be considered statistically significant. Simple bivariate analyses will be conducted to establish initial associations between potential predictor variables and relevant outcome variables. To investigate the associations between pain and lower extremity kinematic and alignment variables, we will use multilevel multivariable logistic regression models, stratified by grade (1st, 5th, and 9th). Examiner ID will be included as a random factor in the models. The statistical analysis will be performed using the newest available version of R [39].

### 6.4. Test and Re-Test Analysis and Sample Size

Initially, the examiners were instructed by an experienced pediatric orthopedic consultant in the full examination program, and a pilot evaluation with three formal tests of five representative, typically developed children with the full evaluation program was performed by all examiners at least a week apart as an initial evaluation of the intra- and inter-variability [40,41]. The results were performed to evaluate each parameter to be evaluated with a goniometer or as a binary parameter as well as how these tests should be performed. The examiners were then re-instructed in the examination program. A test and re-test analysis was carried out in the spring of 2023 with a subset of 50 children. This sample size is often recommended for test–retest studies [42]. Multiple examiners independently assessed these children at two different time points less than 4 weeks apart. To quantify the reliability of our assessments, we will calculate reliability coefficients using intra-class correlations (ICC 3.1) [43]. Additionally, we will express absolute measurement errors using limits of agreement [44]. To enhance the clinical relevance of our study, we will engage three chief orthopedic surgeons to provide insights into acceptable measurement errors in our context. Each surgeon will independently assess what they deem as clinically acceptable thresholds for measurement errors. Should there be any discrepancies in their assessments, a joint meeting will be organized to discuss and reach a consensus. The agreed-upon acceptable thresholds will guide our interpretation of the limits of the agreement.

## 7. Expected Results

The focus on children’s health is essential for children, caregivers, and healthcare professionals such as school nurses, doctors, and podiatrists [1,2]. The lower extremities and feet of typically developed children are an area that causes pain [5]. Specific sports activities seem to impact the health of Danish children’s diseases of the lower extremities and feet, leading to pain [45,46], which, subsequently, will lead to activity restrictions and, if causing musculoskeletal overuse injury, healthcare service needs. To improve the care of children’s feet, a project for the evaluation and screening of children’s feet and lower extremities in Danish primary schools was initiated in the fall of 2022. The interdisciplinary project between physicians, researchers, and podiatrists will examine the extent of lower extremity and foot diseases in children by evaluating children in the first, fifth, and ninth grades of primary school. The project aims to identify areas of improvement in the care of children’s lower extremities and feet with the tenets of creating a prospective database using online submitted PROMs for further follow-up during adolescence and into adulthood and of providing recommendations for preventive actions for healthcare professionals. The study design is a cross-sectional examination of a thousand children, analyzing their clinical status, pain levels, sports activities, foot dimensions and pressures, and footwear. This study will measure foot dimensions using commercial 3D scanning techniques and assess the fitting of footwear. This project will help provide insights into the health of children’s feet and potential mismatches between foot size and footwear that could lead to deteriorating foot health. This project’s findings aim at developing recommendations concerning anthropometric parameters that might be preconditions for pain, the three-dimensional foot dimensions and foot pressure mapping, and the intensity and type of sports activity as a potential preventive measure from the perspective of socioeconomic inequality on a group level and activity status with the tenet of minimizing unintended consequences, such as performing specific high-intensity sports if preconditioned for pain, for Danish children’s foot health. However, as eluded earlier, the need for elucidating these issues is national, but these issues relate to children worldwide, and this project could be expanded internationally since the clinical and instrumental evaluations, commonplace clinical examinations, and PROMs are translated into several languages; thus, this project would be ‘translational across countries’.

This study is expected to provide information on misalignments and malpositions, deformities, gait patterns, footwear, and skin and foot hygiene in the lower extremities and feet of Danish children, and this will be examined from the perspective of detecting and clarifying the role of sports activities and socioeconomic inequalities on a group level. When analyzing and interpreting data in this study and drawing conclusions thereof, the generalizability will rely upon the specificity of the evaluation methods and sample size. In this study, we will perform a test–retest study and subsequent power analyses to evaluate this. We foresee that we will identify subgroups with specifically located pain [45,46]; thus, we will have to be aware of generalizing small subgroups to the general pediatric population. Therefore, unusual, single, or unexpected results must be carefully considered concerning the profile of other results and their plausibility. Nonetheless, the results obtained in this study may have an impact on public health concerning physical (sports) activity, the mapping of podiatric diseases, and anthropometric parameters in children [47,48].

In conclusion, this study will focus on children’s lower extremities and feet to evaluate the relationship between pain status and physical (sports) activity and the mapping of podiatric diseases. Moreover, this will be examined in the context of socioeconomic status on a group level, foot health, and quality of life. This project evaluating children’s feet and lower extremities in Danish primary schools is a step toward identifying areas of improvement in the care of children’s feet and providing recommendations for parents and healthcare professionals. Overall, this study can potentially have an impact on public health concerning physical activity status and anthropometric parameters in children, which can be the first step in improving children’s foot health and quality of life [48,49].

## Figures and Tables

**Figure 1 mps-06-00115-f001:**
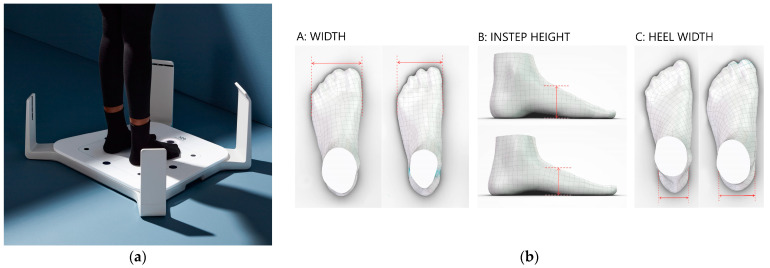
The three-dimensional foot scans: (**a**) the Volumental 3D foot scanner; (**b**) the 3D foot scans and measured foot dimensions.

**Figure 2 mps-06-00115-f002:**
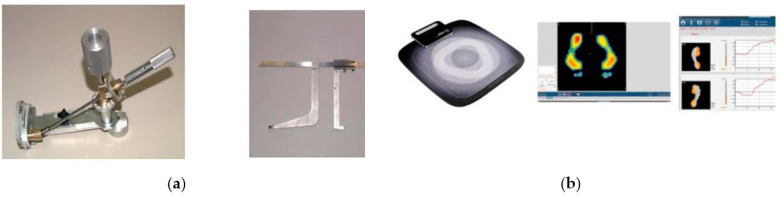
Other instrumental foot-related measurements: (**a**) the ISC In-Shoe Length Measuring Device and In-Shoe Width Measuring Device; (**b**) the Sidas foot pressure scanner and the foot pressure mapping.

## Data Availability

There are no data available.

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
