# Peer review of "Protocol Article: A Cross-Sectional Evaluation of Children’s Feet and Lower Extremities"

_mps, 2023, doi:10.3390/mps6060115_

Round 1

Reviewer 1 Report

Comments and Suggestions for Authors

Dear authors

The paper has an overall aim that is very interesting. However the aims and research questions need to be clarified. An ethical approval is needed and ethical considerations need to be described in the paper. 

Detailed comments and suggestions for improvements are found in the attached documents. 

I have read the appendices but can not upload additional files to you. My main comments was: 

The common way to express is that you "ask" participants to participate you do not invite them to a study. Follow the ethcial regulations in Denmark.

Comments on the Quality of English Language

N/a

Author Response

Dear reviewer

Thank you for your good and thorough review. We agree with your suggestions and have modified the manuscript accordingly in the submitted document.

When we reference to a line number from the previously submitted manuscript, the line number refers to the previously submitted manuscript, and when writing about our suggestions for the revised manuscript, the line number refers to the current version of the submitted manuscript.

This response is structured to answer your specific suggestions and these are also included in the revised manuscript using italics.

Thank you

Warmest regards

The corresponding author

Reviewer 2 Report

Comments and Suggestions for Authors

Very important and huge project on pediatric foot. Well planned. 

1. The main question of this project is how to broaden the knowledge of the pediatric foot  - disorders, p[roblems, and then how to react. Very important topic, Up to now we know only some pieces of the whole. 2. The topic is very important - it's original and relevant, especially due to its holistic attitude to foot, shoes, etc.We need such a description for better prophylaxis and treatment in foot problems. 3. I am waiting for the results - they update my knowledge on pediatric foot - the influence of modern shoes, obesity, etc.  4. I do not think that any improvements are necessary - maybe only a clear info campaign before start - to encourage teenagers to participate in the project 5. conclusions are OK. 6. references are appropriate 7. a new figure with advertisement poster for schools may be beneficial  

Author Response

Dear reviewer

Thank you for your good and thorough review. We thank the reviewer for the remarks and agree with your suggestions and have modified the manuscript accordingly.

This response is structured to answer your specific suggestions and these are also included in the revised manuscript using italics.

Thank you

Warmest regards

The corresponding author

Reviewer 3 Report

Comments and Suggestions for Authors

I have read your article with interest and find it covers an important topic. However, to strengthen the manuscript and project protocol I would recommend you consider some revisions. Specifically:

Introduction

Provide more context on why this study is important - what are the potential benefits to children's health and quality of life? Explain in more detail how this study could lead to preventive strategies.

Clarify the rationale for specifically selecting children in grades 1, 5, and 9. Do they represent key developmental milestones? Do they represent key developmental milestones?

Briefly describe relevant previous studies in this area. What gaps in the existing literature does this study seek to fill?

Methods

Explain in detail how the sample size of 1000 children was determined. Was any statistical analysis performed to calculate the number needed for adequate power?

Provide more detail on the inclusion process - how will schools and participants be selected and recruited? 

Explain how inter-rater inter-rater reliability will be established - will a test-retest analysis be performed?

Describe the statistical analysis plan more clearly - what tests will be used for what types of data? How will confounding factors be accounted for?

It is convenient to put that the clinical study you want to perform is a clinical, observational or epidemiological study, since you will not be performing any intervention.

You should justify each of the variables of the project, such as, for example, for the measurement of the subtalar role in degrees and as you will perform it this way with goniometer, the same force will be applied to all. If the operator will be the same, if there will be several operators, how will you avoid bias in the measurements?

The type of foot should justify if there are only Roman or Greek, or it could also be square. 

The figures are not observed

The methodology described in section 3.1 as it says in line 256 does not describe how you will collect the information.

Results

Elaborate on the potential public health impact with respect to factors such as socioeconomic status and physical activity.

Discussion

Emphasize the prospective database this could create for longitudinal follow-up of children over time - how will children be followed?

Mention possible generalizability issues - will the results apply to other populations outside of Danish children?

In general, provide further methodological details, justifications and discussion of limitations and implications to strengthen the protocol.

In the concluding statements you should fill in each of the items again.

With these changes, I believe that the project would be improved and seriously extrapolated to other countries. I am happy to discuss these recommendations in more detail if you wish. Thank you for the opportunity to review this paper. Sincerely

Author Response

(The authors gave the same response as above.)
